# Immune System and Hepatocellular Carcinoma (HCC): New Insights into HCC Progression

**DOI:** 10.3390/ijms241411471

**Published:** 2023-07-14

**Authors:** Maria Kotsari, Vassiliki Dimopoulou, John Koskinas, Athanasios Armakolas

**Affiliations:** 1Physiology Laboratory, Medical School, National and Kapodistrian University of Athens, 11527 Athens, Greece; mariakotsari@med.uoa.gr (M.K.); vdimopoulou14@gmail.com (V.D.); 2B’ Department of Medicine, Hippokration Hospital, National and Kapodistrian University of Athens, 11527 Athens, Greece; jkoskinas@med.uoa.gr

**Keywords:** hepatocellular carcinoma, tumor microenvironment, tumor-associated macrophages, immunotherapy

## Abstract

According to the WHO’s recently released worldwide cancer data for 2020, liver cancer ranks sixth in morbidity and third in mortality among all malignancies. Hepatocellular carcinoma (HCC), the most common kind of liver cancer, accounts approximately for 80% of all primary liver malignancies and is one of the leading causes of death globally. The intractable tumor microenvironment plays an important role in the development and progression of HCC and is one of three major unresolved issues in clinical practice (cancer recurrence, fatal metastasis, and the refractory tumor microenvironment). Despite significant advances, improved molecular and cellular characterization of the tumor microenvironment is still required since it plays an important role in the genesis and progression of HCC. The purpose of this review is to present an overview of the HCC immune microenvironment, distinct cellular constituents, current therapies, and potential immunotherapy methods.

## 1. Introduction

Liver cancer is the sixth most common cancer worldwide in terms of incidence rate with approximately 906,000 cases reported annually and is the third most lethal form of cancer, with a five-year survival rate of 18% [1]. Recent evidence suggests that liver cancer incidence, as well as the death rate, increased by at least 43% from 2000 to 2016 (from 7.2 to 10.3 deaths per 100,000) [2]. Hepatocellular carcinoma (HCC) accounts for 75–85% of cases [1]. The majority of HCCs occur in patients with underlying liver disease, usually as a result of chronic hepatitis B (the major risk factor, accounting for 50% of HCC cases) or C, virus infection (HBV or HCV), alcohol abuse, and metabolic syndrome [3]. HBV and HCV infections are major global public health problems with an estimated 1 million and 450,000 deaths yearly worldwide, respectively [4].

Although HBV vaccination and HCV antiviral therapies have reduced HCC occurrence, HCC incidence continues to grow, mainly because of hazardous alcohol use and obesity in western countries [5]. The pathophysiology of HCC is a complex multistep process, with heterogeneous histological features and a diverse mutational landscape [6,7,8,9]. In 80% of HCC cases, telomerase presents activation, which is often induced by telomerase reverse transcriptase promoter mutations [7,10].

Chronic liver diseases are associated with the induction of inflammatory signals, leading to homeostatic imbalance, which is associated with necroinflammation [11]. During chronic HBV infection, the load of circulating HBV or HBV-derived antigens promote T-cell exhaustion, leading to inactivity and subsequent death. Therefore, these individuals present with weaker immunity [12,13,14]. Chronic HCV infection is capable of avoiding immune system recognition due to the high mutational rate [15] and through viral factors that counteract DNA sensors [16,17,18]. In the case of chronic alcohol consumption and non-alcoholic fatty liver disease (NAFLD), sterile inflammation is evident due to cell stress and lipotoxicity. This condition leads to the induction of pro-inflammatory signals and to the activation of monocytes, macrophages, and neutrophils [19,20].

In addition, solid tumors of the liver consist of malignant tumor cells and stromal cells [21,22]. Recent evidence indicated that the stromal cells play a major role in tumor development and progression [22]. Tumor-associated macrophages (TAM), neutrophils (TAN), and dendritic cells are important components of the tumor microenvironment (TME) and can promote tumor progression (proliferation, metastases, invasion). The suppression of the immune system and especially of the T-cells that are observed in chronic liver disease is associated with the development of HCC [23].

HCC is characterized by wide heterogeneity. Recent sophisticated broad range analysis techniques of nucleic acids (next generation sequencing) or proteomics have contributed to the identification of many candidate genes that may be responsible for HCC development mainly affecting cell-cycle control, the Wnt/β-catenin pathway, and epigenetic machinery [24,25]. Some of them have been proven beneficial targets in the therapy of HCC. Additionally, investigations of the microenvironment showed its potentiality to induce the development and growth of HCC by providing a safe niche for the cancer cells resulting from the downregulation of immune system activity. The tumor microenvironment is believed to present a more uniform gene expression pattern amongst patients rather than HCC, rendering it an attractive target for novel therapies against HCC. In the current setting, the new immunotherapy regimens increased the medial survival of HCC patients with intermediate or advanced HCC from 3 months with tyrosine kinase inhibitors (TKIs) to 20 months, although hepatic resection and liver transplantation are the main curative treatments [26]. Therefore, targeting of the TME is more feasible and may provide a more efficient solution to the treatment of HCC [27].

Despite the immense achievements, enhanced molecular and cellular characterization of the tumor microenvironment is necessary since it plays a key role in the origin and development of HCC. This review expands on the important TME components and describes the immune system’s participation in disease progression. Furthermore, emphasis will be placed on the intrinsic mechanisms of immunosuppressive action in the HCC tumor microenvironment, the complex intracellular communication network, and the therapeutic strategies targeting them. Both current knowledge regarding possible treatments and feasible therapies that target the tumor microenvironment will be examined.

## 2. Microenvironment in Cancer

Neutrophils, monocytes, resident macrophages (Kupffer cells [KCs]), natural killer (NK) cells, natural killer T (NKT) cells, and liver-transiting and/or resident lymphocytes (B, CD8+ T, CD4+ T, and T-cells) are all found in the liver [11]. To maintain global homeostasis, the liver environment is strongly tolerogenic against gut-derived microbial metabolites [28,29]. This immunotolerance caused by continuous antigen presentation between liver-resident cells (hepatocytes, endothelial cells, KCs, and dendritic cells [DCs]) and peripheral leukocytes in the absence of costimulatory molecules, allowing regulatory T-cells (Treg) to expand in response to KC-derived IL-10 [30]. There is an overall balance between anti-inflammatory cytokines (IL-10, IL-13, and transforming growth factor β [TGF-β]) and proinflammatory cytokines (IL-2, IL-7, IL-12, IL-15, and interferon γ [IFN-γ]), which maintains homeostasis [31].

The precancerous microenvironment (PME) creates the circumstances for carcinogenesis prior to the establishment of the TME. PME is a microenvironment that occurs prior to tumor formation and is characterized by persistent liver damage, inflammation, and fibrosis [32]. The ‘liver disease trilogy’ consists of hepatitis, cirrhosis, and liver cancer. Surprisingly, fibrosis is a post-tumor development reaction in all cancers except HCC [33]. As a result, PME is an early warning sign of carcinogenesis, and identifying its specific markers could assist in predicting the incidence of HCC. The precancerous microenvironment gradually converts into the tumor microenvironment as the tumor grows.

More precisely, the tumor immune microenvironment (TIME), which serves as a base for interactions between tumor cells and immune cells, is critical to HCC progression and has a significant impact on immunotherapy outcomes. Many immune cells have been discovered to aggregate during tumor growth within the TIME, including myeloid-derived suppressor cells (MDSCs), regulatory T (Treg) cells, and tumor-associated macrophages (TAMs), which are responsible for the formation of an immunosuppressive environment. NK cells, cytotoxic CD8+ T cells, and CD4+ T cells with a proinflammatory T-helper 1 phenotype, on the other hand, collaborate to prevent protumor effects [34]. Recently, TIMEs cells were classified into three types based on the degree of immune infiltrate: infiltrated-excluded (I-E) TIMEs, infiltrated-inflamed (I-I) TIMEs, and tertiary lymphoid structure (TLS) TIMEs [35]. I-E TIMEs are densely packed with immune cells but devoid of cytotoxic lymphocytes (CTLs) in the tumor core, whereas I-I TIMEs are densely packed with CTLs expressing programmed cell death 1 (PD-1), and leukocytes and tumor cells within I-I TIMEs express immune-dampening PD-1 ligand (PD-L1). TLS-TIMEs are a subclass of I-I TIMEs that include TLSs with lymphoid aggregates resampling cell composition similar to that of lymph nodes. These immune content classifications within the TME are the leading information for distinguishing different immunological compositions and the immune status of tumors. For example, tumors with an I-E TIME are characterized as immunologically “hot” tumors, which are responsive to immunotherapy and are associated with better overall survival [36].

### 2.1. Neutrophils

Neutrophils are innate immune cells that infiltrate a tissue upon infection, damage, or cancer. Tumor-associated neutrophils (TANs) have been found to correlate with tumor growth, lymph node metastasis, and poor prognosis in various solid human malignancies [37]. TANs, on the other hand, occur in two varieties: antitumorigenic (N1) and protumorigenic (N2). The difference between N1 and N2 phenotypes depends on different levels of activation (by TGF-β) rather than different molecules [38]. Protumorigenic N2 TANs can create decondensed chromatin embedded with granular proteins, known as neutrophil extracellular traps (NETs), which are known to promote tumor development [39]. CD66b+ neutrophils concentrated in the peritumoral region are associated with a lower overall survival in HCC [40].

As essential effectors in the fight against cancer, neutrophils account for a significant proportion of leukocytes in the circulatory system [41]. These neutrophils fight cancer in a variety of ways, including antibody-dependent cellular cytotoxicity (ADCC), direct cytotoxicity, and the activation of antitumor adaptive immunity [42,43]. In addition, N1 TANs have been linked to enhanced NADPH oxidase activity, which results in the generation of ROS, which are known to be lethal to tumor cells [44,45]. TAN-produced chemokines and pro-inflammatory cytokines, including chemokine (C single-bond C motif) ligand 3 (CCL3), chemokine (C single-bond C motif) ligand 9 (CCL9), chemokine C-X-C ligand 10 (CXCL10), tumor necrosis factor TNF-α, and interleukin 12 (IL-12), can recruit and activate CD8+ T-cells, which can limit tumor growth [46]. TANs have also been shown to interact with macrophages and induce IL-12 production, which contributes to the polarization of type 1 CD4- CD8- unconventional T-cells αβ (UTC_β_). Furthermore, such Type 1 UTC_αβ_ cells could release IFN-γ for anti-tumor immunity in a variety of malignancies [47].

Neutrophils can undergo phenotypic and functional modification driven by tumors and TMEs, which could promote tumor growth via a variety of processes. Neutrophils are mildly activated into TANs in the TME, which then produce chemicals including ROS and neutrophil elastase (NE) that promote tumor development and invasion. Aggregated neutrophils can also create OSM when monocyte-derived TNF and tumor-derived soluble factors interact, which may contribute to cancer spread and disease progression in HCC [48]. TAN is defined by increased PD-L1 expression, which can impair T-cell immunological activity in HCC [49]. After neutrophil activation, DNA, histone, and granzyme components, such as neutrophil elastin and myeloperoxidase, are released into the cell, forming a network structure that has been shown to prevent tumor cells from contacting cytotoxic T lymphocytes and CD8+ T-cells by overlaying them [50]. Furthermore, NETs cause tumorous inflammation by activating the toll-like receptor TLR4/9-COX2 axis, which promotes tumor metastasis [51]. Neutrophil counts were found to be positively linked with MDSC levels, which are well-known promoters of cancer. In the experimental setting, TANs producing CCL2 and CCL17 chemokines were observed to recruit TAMs and Foxp3+ Tregs and increase HCC growth (Figure 1) [52]. Finally, cancer-associated fibroblasts (CAFs) were discovered to upregulate PD-L1^+^ neutrophils via the IL-6/STAT3 axis, compromising T-cell activity via the PD-L1/PD-1 signaling pathway, which is required for neutrophil survival and functional activation (Figure 1) [53].

### 2.2. Macrophages

As one of the most common types of innate immune cells, macrophages act as the body’s initial line of defense against pathogenic insults. Macrophages are prevalent in all tissues and have a high level of plasticity and functional variety [54]. Macrophages participate in phagocytosis, antigen processing and presentation, and immune system organization through the production of various cytokines, which regulate the beginning of inflammation, progression, and resolution [55,56].

The liver hosts the majority of the body’s macrophages and is overseen by myeloid cells such as blood monocytes, which scan the hepatic vasculature and eventually infiltrate into the liver. In homeostasis, monocyte-derived cells can evolve into liver DCs or monocyte-derived macrophages (MoMFs), which contribute to the pool of resident macrophages known as Kupffer cells. Kupffer cells develop throughout embryogenesis from yolk sac-derived progenitors, establishing a self-renewing pool of resident macrophages in the liver and serving critical functions in maintaining hepatic and systemic homeostasis [57]. In invasion conditions, Kupffer cells are activated and can cause chronic liver inflammation by recruiting immune cells to the liver, particularly monocytes, from which DCs and MoMFs are produced. Until now, no particular markers have been demonstrated to differentiate human Kupffer cells from monocyte-derived cells [58].

Macrophages are classified into two distinct categories based on their environmental stimuli: M1 and M2. M1 macrophages are activated by microbial components or proinflammatory cytokines (TNF, IFN-γ, TLR) (Figure 1) and trigger proinflammatory activities such as the release of nitric oxide, reactive oxygen species, and the proinflammatory cytokines IL-1, IL-6, IL-12, TNF-, CXCL5, and CXCL8/10. M2 macrophages, on the other hand, are polarized by IL-4, IL-10, IL-13, and glucocorticoids (Figure 1) and perform immunosuppressive roles to promote tissue healing. The CD68 marker is usually utilized to locate and distribute liver TAM, and the expression levels of CD86 (M1), CD163 (M2), and CD206 (M2) are regarded as sufficient to differentiate M1 and M2 macrophages in vitro [59]. However, the M1/M2 terminology has become contentious due to the existence of numerous unique polarization phenotypes seen in tissues and caused by a variety of diverse stimuli [57,60]. There are various subtypes of the M2 subclass (M2a, M2b, M2c, and M2d) that are distinguished by their activation triggers [61]. However, because of the large range of activation states and indicators, these subclasses are difficult to be identified in vivo [62].

Since TAMs have similar molecular and functional profiles, the M2 phenotype appears to be the dominant macrophage phenotype in malignancies. These profiles are defined by low levels of differentiation-associated macrophage markers including carboxypeptidase M and CD51, high constitutive production of arginase I, IL-1 decoy, IL-1ra, IL-6, and IL-10, and low expression of TNF and IL-12 [63].

Plenty of chemokines (CCL2, CCL5, CCL15, CCL20), cytokines (such as CSF-1),and complement cascade products have been shown to have a role in the process of monocyte-derived macrophage recruitment and migration [64]. Furthermore, some studies have recently shown evidence of the transition from Kupffer cells to the TAM pool by activating the Her2/Neu pathway [65,66]. Other factors implicated in TAM recruitment include mitochondrial DNA (mtDNA), osteopontin (OPN), microRNAs (miRNAs), circular RNAs (circRNAs), and HCC cell-derived exosomes [67,68,69,70]. TAMs promote mutual epithelial cell-macrophage dependence in HCC by inducing ST18 expression in epithelial cells [71]. TAMs also produce immunosuppressive chemokines and cytokines, such as IL-10 and TGF-β, which play immunoregulatory roles. TAMs have also been found to recruit Tregs to the tumor (Figure 1), which compromises the activation and function of effector T-cells [72]. Recent research has shown that insulin-like growth factors (IGF)-1 and IGF-2 modify macrophages during maturation [73] and the tricarboxylic acid cycle metabolism is important in macrophage epigenetic reprogramming [74].

Furthermore, macrophages have a variety of tumor-promoting roles, including immune suppression, metastasis, angiogenesis, cancer cell stemness maintenance, and treatment resistance [75,76]. According to many studies, high levels of TAMs, particularly in the peritumoral area, are associated with a poor prognosis in individuals with HCC [77,78]. TAM infiltration within the tumor or at the margins can also predict a poor outcome following tumor excision [79]. The study of CD68+-TAMs in a cirrhotic HBV-positive cohort of 137 patients revealed that marginal macrophage density was related to vascular invasion, tumor multiplicity, and fibrous capsule formation. In an HBV^+^-related cirrhosis/HCC cohort (*n* = 253), a dysregulated balance toward CD206+ M2 macrophages was linked with an aggressive phenotype with advanced tumor-node-metastasis stage, poor overall survival, and shorter time to recurrence [80]. In a small group of patients (*n* = 26), the density of TAMs was similarly linked to resistance to trans arterial chemoembolization [81].

### 2.3. Monocytes Myeloid-Derived Suppressor Cells (MDSCs)

Monocytes are innate immune cells with a dual involvement in cancer. They are frequently recruited into tumors via tumoral CCL2 production [82]. Various subsets of monocytes have been identified in HCC as the tumorigenic process progresses. Monocytes can be divided into three subtypes: CD14+, CCR1 + CD14+, and myeloid-derived suppressor cells (MDSCs). With the production of immune-checkpoint inhibitors (PD-L1/2, B7-H3, TIM3) and cytokines (IL-10, CXCL2, CXCL8), all of these subtypes produce a strong immunosuppressive environment, inhibiting NK cytotoxicity and activating Tregs (Figure 1). They also collaborate with neutrophils to increase tumor invasiveness via the oncostatin M pathway [83]. Recruited monocytes can kill tumor cells in the early stages. However, tumors that escape immune monitoring thwart monocyte-induced death by reprogramming the monocytes into immune-suppressive cells [84]. CCR1+CD14+ tumor-educated monocytes express PD-L1, B7-H3, and TIM3, upregulate tolerogenic metabolic enzymes, and induce angiogenesis and metastasis [85].

MDSCs are a diverse population of myeloid cells that have been found to decrease T-cell responses in cancer and HCC. There are two types of MDSC populations: monocytic (M-MDSC-CD14+) and polymorphonuclear (PMN-MDSC-CD11b + CD33 + HLA-DR-). Both share phenotypic and morphologic characteristics with monocytes and neutrophils, respectively [86]. In general, IFN-γ, toll-like receptor ligands (TRL), IL-13, IL-4, and TNF-β all activate MDSCs in TME [87]. In HCC, MDSCs primarily decrease T-cells’ immune responses. MDSCs in TME have enhanced arginase activity and can utilize the cysteine of antigen-presenting cells competitively. Thus, by consuming arginine-l (Arg-1) and L-cysteine in the microenvironment, MDSCs impair T-cell activity [88].

MDSCs produce unique cytokines, such as Galectin-9, which binds to TIM-3 on T-cells and causes T-cells apoptosis [89]. A3B expression was shown to be increased in HCC patients, although no apolipoprotein B mRNA editing catalytic polypeptide (APOBEC) mutation pattern was discovered. The mechanism study revealed that upregulated A3B inhibited the enrichment of spherical H3K27me3 by interacting with multi-comb repressor complex 2, reducing H3K2me3 binding to the CCL2 promoter, activating CCL2 transcription in HCC cells, and recruiting a large number of MDSCs into the HCC microenvironment [90]. In addition, MDSCs from HCC patients can decrease the cytotoxicity and IFN-γ release of autologous NK cells by direct interaction, mostly via the NKp30 receptor on the NK cell surface [91]. CAFs can also stimulate MDSC production, recruiting monocytes via the SDF1a/CXCR4 pathway and promoting differentiation into MDSCs via IL-6-mediated STAT3 activation (Figure 1) [92]. In addition to promoting tumor angiogenesis, tumor-associated endothelial cells (TEC) can produce IL-6 through the NF-B signaling pathway and recruit MDSCs in order to assist tumor cells to avoid immune clearance [93].

### 2.4. Dendritic Cells (DCs)

DCs represent an important connection between innate and adaptive immunity since they regulate antigen presentation, which leads to T-cell activation and differentiation. Unlike macrophages, DCs migrate and present antigen to T-cells in tissue-draining lymph nodes [94]. Although many DC–T cell interactions also take place in the liver [95]. The activation of CD8+ T cells is dependent on the activation of a DC by CD4+ T-helper (Th) cells (Figure 1) [96].

Three controlled processes define a fully functional immunological synapse. Firstly, DCs must exhibit the antigen to CD4+ T cells via MHC class II molecules and CD8+ T cells via MHC class I molecules. Second, costimulatory molecules from the immunoglobulin superfamily (CD80-CD86/CD28) and the TNF superfamily (CD40L/CD40, 4-1BBL/4-1BB, CD27/CD70, CD30L/CD30, and HVEM/LIGHT) must interact to stimulate the production of cytokines that promote CD8+ T cell expansion and differentiation (third step). CD4+ Th cells (“classical licensing”) or NKT cells (“alternative licensing”) can assist with CD8+ T-cell licensing [97]. Interference with any of these three phases will result in a faulty adaptive response. As a result, one of the primary strategies by which cancer cells avoid immune monitoring is the disruption of this immunological synapse, which is frequently accomplished by the production of inhibitory ligands that limit T-cell activation [98]. T-cell exhaustion is described as decreased T-cell capability to proliferate and produce cytokines, which can be induced by overexpression of inhibitory immune-checkpoint receptors such as PD-1, CTLA4; lymphocyte-activating 3 (LAG3),hepatitis A virus cellular receptor 2, and TIM3. In general, inhibitory immune-checkpoint ligands are expressed not exclusively by HCC tumor cells but additionally by myeloid cells including DCs, TAMs, and neutrophils [99].

Human DCs frequently display the MHC molecule, human leukocyte antigen (HLA)-DR, cluster of differentiation (CD) 209, and integrin CD11c, which are also seen on macrophages, activated T-cells, B-cells, and natural killer (NK) cells. DC populations vary greatly depending on developmental lineage, differentiation period, and the impact of surrounding tissue [100]. The combination of the four markers, BDCA-1 (CD1c), BDCA-2 (CD303), BDCA-3 (CD141), and BDCA-4 (CD304), defines the primary subsets of DC. Conventional DCs (cDCs), previously referred to as myeloid DCs, are classified as CD141+/CD14-type 1 cDCs (cDC1) and CD1c+/CD14-type 2 cDCs (cDC2). The latter are the most common kind of DC in the human liver but are not frequently observed in peripheral blood or the spleen [101]. Plasmacytoid DCs (pDCs) produce CD303 and CD304, organize antiviral responses, and release type 1 interferon (IFN), which stimulates NK cells, B cells, T-cells, and myeloid DCs [102]. Monocytes can differentiate into inflammatory DCs (infDCs) in persistently inflammatory circumstances, a subpopulation that can stimulate T-helper cell (Th) 17 cell differentiation from naive CD4+ T-cells. InfDC development shares traits of both DC and macrophage development, as well as aspects of in vitro-generated, monocyte-derived DCs (moDCs) [103]. Single-cell RNA sequencing and single-cell protein analyses have enhanced our knowledge of DC variation by indicating that the cDC2 subtype can be additional subdivided into different categories, consisting of a circulating inflammatory subset and allowing a clear distinction between monocytes and cDC2 [104].

### 2.5. Natural Killer Cells (NKs)

NK cells are granular lymphocytes that initiate direct innate immune responses against infections and cancer cells [105]. Human NK cells represent around 15% of all lymphocytes and are phenotypically characterized by the presence of CD56 expression with the lack of CD3 expression, along with more than 30% of NK cells identified primarily in the liver [106].

NKs have been separated into two major populations: immune-modulator (CD56bright/CD16) and cytotoxic (CD56dim/CD16+). CD56dim NK cells account for roughly 90% of all blood NK cells and exhibit high quantities of the low-affinity Fc receptors CD16 and killer immunoglobulin-like receptors (KIRs), whereas immune-modulator NK cells account for less than 10% of all blood NK cells and do not express CD16 and KIRs [107]. CD56dim NK cells efficiently kill target cells by degranulation but produce minimal amounts of cytokines when stimulated. On the contrary, CD56bright NK cells release high quantities of cytokines when activated but are less cytotoxic. CD56bright cells, on the other hand, displayed comparable or greater cytotoxicity to target cells following sustained activation in comparison to CD56dim cells [108]. Furthermore, CD56bright NK cells express high- and intermediate- affinity interleukin 2 receptors (IL-2R) and respond in vitro and in vivo to low doses of IL-2 [109].

Inhibitory receptors, such as inhibitory killer immunoglobulin-like receptors and the C-type lectin-like receptor NKG2A, which bind MHC-I and the nonclassical MHC-I complex, HLA-E, are expressed on NK cells. When these ligands are not present on a cell, activating NK receptors (NKp30, NKp46, NKp44, NKG2D, and NKG2C) bind their potential ligands produced on infected or tumor-transformed cells [105]. The enhanced cytotoxic ability of peripheral blood NK cells is associated with a lower risk of cancer [110]. In addition, a large number of NK and CD8+ T lymphocytes predicts a better prognosis in the early stages of human HCC and is associated positively with apoptotic tumor cells [111]. Despite this, NK cells often lose their anticancer characteristics. In patients with stage III HCC (*n* = 110), the quantity of both peripheral blood and liver CD56dim NK cells is reduced, although immune-modulator NKs are increased [112].

### 2.6. Cancer-Associated Fibroblasts (CAFs)

CAFs are a significant feature of the HCC microenvironment, and the predominant types of CAF observed in certain malignancies are activated fibroblasts [113]. According to emerging data, CAFs are a diverse group of cells that rely on their various biological precursors [114]. CAFs may be identified by a variety of markers, including SMA, fibroblast activation proteins (FAPs), fibroblast specific protein 1 (FSP1), vimentin, and PDGF receptors (PDGFRs)-α and β [115,116]. Even so, the lack of fibroblast-specific markers makes pinpointing the particular cells of origin of CAFs difficult. According to recent studies, CAFs can be derived from pancreatic and hepatic stellate cells (HSCs), neutrophils, bone marrow-derived mesenchymal stem cells (MSCs), adipocytes, pericytes, endothelial cells, and cancer cells undergoing epithelial-mesenchymal transition (EMT) [117]. However, the bulk of these CAF precursor cell types have been found through in vitro studies and bone marrow transplantation procedures [118].

The number of CAF studies has expanded substantially in the recent decade, reflecting the fact that CAFs represent the key element of the stromal cell population in a TME. CAFs are frequently involved in the evolution of HCC and treatment resistance [119]. They influence HCC growth through a variety of methods, including direct impacts on HCC cells via soluble factors and exosome release and indirect effects via other stromal cells and ECM remodeling [120]. CAFs promote cancer development and metastasis by producing a number of soluble substances, including inflammatory cytokines, growth factors, and chemokines [121,122,123,124]. CAFs are phenotypically and functionally diverse, capable of both protumorigenic and antitumorigenic activity [125].

Previous research has linked CAFs to HCC cell metastatic growth and invasion; however, the mechanisms by which CAFs induce HCC metastasis have not been entirely defined [126]. In some circumstances, activation of stem cell signaling has been associated with resistance to CAF-mediated treatment. A soft matrix stiffness enhances tumor development by allowing tumor cells to spread out. As HCC advances, a stiffer ECM promotes cancer stem cell (CSC) proliferation and self-renewal, while soft a ECM may assist CSC metastasis [127]. CAFs do not reside in isolation surrounding tumors;instead, they interact with tumor cells to enhance their malignant characteristics [128]. CAF precursors can be recruited by tumor cells, and normal fibroblasts can be transdifferentiated into CAFs. At the same time, CAFs produce a substantial number of cytokines, chemokines, growth factors, and ECM proteins, which combine to generate TMEs, which increase HCC cell proliferation, metastasis, and treatment resistance. The chemokine-chemokine receptor (CK-CKR) network controls immune cell recruitment and affects the TME [129]. CAFs have been shown in studies to increase CCL2, CCL5, CCL7, CCL26, and CXCL17 levels and to adopt a promoter-tumor-metastatic nature [124].

### 2.7. T-Cells

The composition of lymphocytes in the liver varies from that of blood and other lymphoid tissues. The liver includes a high number of “unconventional” lymphocytes, such as innate and innate-like lymphocytes, mucosa-associated invariant T-cells, T-cells, NK and NKT cells, as well as T- and B-cells, which are typical constituents of adaptive immunity. In terms of classical T-cells, the human liver has an inverted CD4/CD8 ratio (1:3.5) compared to peripheral blood (2:1) and a higher number of double positive CD3+ CD4+ CD8+ lymphocytes [130]. The hepatic T-cell environment is formed by the attraction of specific lymphocyte forms from the liver sinusoids, where liver sinusoidal endothelial cells (LSECs) produce a substantial quantity of adhesion molecules, such as intercellular adhesion molecule (ICAM) 1, ICAM2, and vascular adhesion protein 1 (VAP1), which interact with activated T-cells, CD8+ T cells (CTLs), and NK cells. Furthermore, the hepatic immune system is enhanced by NKT cells and specific T-cell subsets [131].

CD8+ and CD4+ T lymphocytes are abundant in liver malignancies, specifically within the tumor and in the peritumoral region. Patients with low intratumoral Tregs and a large amount of activated CD8+ T-cells have a better prognosis. Tregs induce T-cell downregulation in HCC (Figure 1), and their presence in the tumor is linked to lower survival rates [132]. However, the involvement of Tregs in chronic liver conditions varies depending on the cause of the disease. While they are more active and increased in viral (HBV, HCV) chronic liver disorders, which supports the chronic infection, they are traditionally decreased in quantity and function in autoimmune liver diseases (autoimmune hepatitis and primary biliary cholangitis). This distinction between viral and autoimmune liver disease may explain why autoimmune liver disease has a lower incidence of HCC than viral chronic liver disease [133]. T-cell heterogeneity was recently studied using scRNAseq in six individuals with HBV+ treatment-naive HCC. Exhausted CD8+ T-cells (PDCD1+CTLA4+) and Tregs (TIGIT+CTLA4+) were abundant in the tumor and expressed the inhibitory marker LAYLIN (LAYN). TCR sequencing found that 10% of CD8+ T-cells in blood and healthy liver tissues carried clonal TCRs, but they reached 30% in carcinomas. Exhausted CD4+ and CD8+ T lymphocytes were found to be more closely related to intermediate populations, producing granzyme A (GZMA) and K (GZMK) markers, than to effector populations, implying prospective treatment techniques that may promote activation over exhaustion [134]. In response to a certain balance of cytokines and chemokines, such as TGF-β and IL-6, distinct CD4+ T-cell populations, that include Th1, Th2, Th17, and Tregs, can be generated [135].

The expression of PD-1 and its ligands PD-L1/2 has been extensively researched in HCC. In normal conditions, their expression serves as a defense response to avoid autoreactive T-cell activation and the death of healthy cells [136]. PD-1 expression is enhanced on CD8+ T effector cells in HCC, and PD-1 interacts with tumors producing PD-L1/2, which suppresses T-cell signaling, proliferation, and cytokine production [137,138]. The percentage of circulating PD-1+CD8+ T cells raised with disease progression in a population of HBV patients [137]. CD8+ T cells trigger PD-L1 expression in hepatoma cells in an IFN-γ-dependent path, which induces the apoptosis of T-cells. HCCs characterized by a defined population of PD-1-high cells are more aggressive, although treatment with anti-PD-1 is expected to be effective [138].

### 2.8. B-Cells

B lymphocytes have many roles in ADCC and antigen presentation, and recent studies suggest that they also play a role in modulating innate and adaptive immunity via cytokine secretion. Despite the fact that B-cells were formerly assumed to be inactive in HCC, the significance of tumor-infiltrating B-cells (TIBs) is still debated [139].

There is a significant amount of data suggesting that B cells can interact with tumor cells directly or indirectly, increasing anti-tumor immunity by enabling other immune processes. In cancer, B-cells activated by tumor cells produce antibodies that aid in anti-tumor immunity, resulting in an effective humoral response [140]. A rise in CD20+ B-cells in the tumor border region is associated with a positive prognosis and is connected to a smaller size of tumor, a lack of vascular invasion, and an increase in CD8+ T lymphocyte concentration, particularly in HBV-associated HCC [141]. Similarly, high levels of B-cell subsets extended survival in two different HCC cohorts [142]. Plasma cells were the most frequent type, indicating that B-cell reactions take place in the TME. Furthermore, plasma cells designated as CD20+ CD79+ cells were found to be strongly related to a better prognosis [143]. Furthermore, the percentage of TIBs was found to be positively associated with the number and activation phase of T and NK cells, as well as decreased tumor cell survival. Actually, the number of TIBs was found to be favorably connected to the percentage of apoptotic tumor cells and inversely related to tumor cell proliferation [144]. Additional studies using immunohistochemistry have revealed that atypical CD20+ memory B cells (IgDIgG+CD27CD38) have a tumor-killing ability and collaborate with CD8+ T-cells which contribute to a better prognosis [141].

B lymphocytes could additionally have a protumoral effect. According to recent research, significant infiltration of CD20+ B lymphocytes into the tumor corresponds with lower rates of differentiation and disease-free survival in HCV-induced HCC patients [145]. B-cells can be associated with pro-tumorigenic processes by activating MDSCs (Figure 1), generating pro-tumorigenic cytokines, and activating immunosuppressive regulatory T-cells [146]. The pro-tumoral pathway is significantly mediated by regulatory B-cells (Bregs), a novel specialized subtype of B-cells. The existing recommended guidelines for identifying Bregs are ambiguous [147]. It has been shown that PD-1 B-cells are the most common subset of Bregs in human HCC and that they promote T-cell dysregulation through an IL10-dependent mechanism (Figure 1), hence promoting tumor growth [148].

## 3. Therapies Targeting Tumor Microenvironment

Surgery (transplantation, resection), local treatment (trans arterial chemoembolization, radiofrequency ablation), and systemic targeted therapy are used to treat HCC according to the stage of HCC, severity of liver disease, and patient performance status [124,125]. However, treatment approaches are curative only for small-sized HCC (two to three centimeters in diameter). Unfortunately, the majority of patients with HCC are already in an intermediate or advanced stage of HCC when diagnosed and systemic treatment is the only option. Recently an increasing number of therapies targeting the tumor immune microenvironment have been developed [149,150].

Recent data suggest that tumor cells with stem cell-like properties are more resistant to conventional therapies than nonstem-like populations. The processes of resistance emergence and the acquisition of stem-like cell features are inextricably related to CSC qualities such as plasticity, quiescence, CSC habitats, and enhanced drug efflux activity [151]. The most common example of plasticity is EMT. Another feature of CSCs that restricts traditional treatment is quiescence. CSCs can briefly enter the Go phase of the cell cycle and remain inactive in the face of microenvironmental changes such as oxidative damage, hypoxia, food restriction, or chemotherapeutic pressure. Most cancer treatments target proliferating cells, and latent CSCs can elude therapy and return to a proliferative state when favorable conditions arise [152]. ATP-binding cassette (ABC) transporters are capable of trading a wide range of toxin-producing chemicals from cells and hence contribute directly to resistance acquisition. CSCs overexpress ABC transporters and dysregulate signaling pathway networks, resulting in multidrug resistance and self-renewal characteristics, accordingly [153].

For this reason, the immunomodulatory function of anti-angiogenic drugs in HCC is also noteworthy. Some TKIs used in HCC therapy have lately been shown to have immunomodulatory effects. Regorafenib, in particular, has demonstrated anti-immunosuppressive properties as well as antitumor immunity promotion by regulating macrophages and boosting proliferation and activation of CD8+ T-cells, and cabozantinib has a complementary effect with immune checkpoint inhibitors by acting on TAMs and reducing tumor vascularity [154]. As a result, current developing data supports the rationale for TKI and ICI combination treatment [155] (Table 1).

### 3.1. Neutrophil-Targeting Therapies

Neutrophils can be used as anticancer therapeutic targets by inhibiting neutrophil recruitment, migration, and activation. An increasing amount of data suggest that neutrophils may play an active role in tumor growth. In human HCC, TANs develop a pro-tumoral N2 phenotype in the mid to late phases of tumor growth, which is associated with elevated CLCF1 levels. It has been suggested that apart from its possible use a predictive biomarker for HCC, specific blocking of CLCF1 signaling may be a potent therapeutic factor for HCC patients [156]. The inhibition of CAFs-induced neutrophil activation by CSF-1 is abolished by CSF-1 antibodies. As a result, combining CAF and TAN therapies can greatly decrease carcinogenesis [157].Additionally, both TGF-β and Axl stimulate CXCL5 production and neutrophil infiltration into HCC tissues, indicating that targeting this axis might be an effective approach against HCC development [158]. However, inhibition of CXCL1 or intercellular adhesion molecule 1 (ICAM-1) expression decreases neutrophil infiltration and promotes liver damage and fibrosis [159]. Furthermore, studies showed that miRNA-223, produced by neutrophils, inhibited HCC progression by targeting multiple inflammatory oncogenes [160]. In conclusion, the role of TANs and the modulation of their function in HCC need further investigation (Table 1).

### 3.2. Macrophage-Targeting Therapies

TAMs appear to play key roles in the genesis and progression of HCC according to recent data. As a result, immunotherapies focusing on TAMs have come to prominence as a viable option for treating HCC patients. Current TAM-targeting protocols include phagocytosis-promoting treatments, monocyte recruitment suppression, excision of pre-existing TAMs in tumor tissue, modifying TAM polarization, and blocking pro-tumorigenic proteins released by TAMs [161].

TAM IL-6 release has been shown to increase CD47 production in HCC cells through the STAT3 signal transduction path. CD47 overexpression has been linked to lower survival rates and recurrence-free survival. In the existence of chemotherapeutic drugs, CD47 inhibition increased TAM-mediated phagocytosis [162]. Moreover, by inhibiting CD47, the HDAC6/let-7i-5p/TSP1 axis decreased the neoplastic and antiphagocytic characteristics of HCC cells, presenting a prospective therapeutic target for HCC therapy [163]. Furthermore, studies showed that the anti-CD47 monoclonal antibody (B6H12) inhibited tumor development and improved chemotherapeutic effectiveness in HCC [164].

Inhibiting monocyte recruitment in HCC tissues has currently been recognized as a viable strategy for lowering TAM levels. A PI3K/AKT pathway-dependent strategy has been reported to reduce HCC development and macrophage infiltration by regulating macrophage colony-stimulating factor (M-CSF) [165]. CCL2 expression has been found to be increased in HCC tissues, and it has been proposed as a new predictive marker for HCC [166]. CCL2 is mainly produced by Kupffer cells and is greatly involved in monocyte-derived macrophage recruitment and modification [167]. CCL2/CCR2 transmission has been identified as a target for suppressing monocyte recruitment in malignancies [168].

Pharmaceutical medicines that inhibit macrophages in vivo, such as clodronate-encapsulated liposomes or amino bisphosphonates, have decreased angiogenesis and tumor growth in various experiments of tumor models. Recent studies indicate that TAMs have a significant effect on tumor progression in the course of sorafenib treatment. Clodronate-encapsulated liposomes and zoledronic acid (ZA), both of which reduce macrophage numbers, are potential medications that, when administered together, increase the anticancer effects of sorafenib [169]. ZA induces apoptosis in particular TAM subsets. Furthermore, ZA therapy has been demonstrated to improve the outcomes of transarterial chemoembolization by inhibiting TAM infiltration in HCC [170].

In HCC-associated macrophages, receptor-interacting protein kinase 3 (RIPK3) is decreased, and RIPK3 inadequacy activates fatty acid oxidation (FAO), which promotes M2-polarized TAMs. As a result, RIPK3 activation or FAO inhibition altered TAM immunosuppressive activity and reduced HCC carcinogenesis [171]. In addition, 8-bromo-7-methoxychrysin (BrMC) was found to reduce M2 macrophage impacts by changing the profile of released cytokines and reversing TAM M2 polarization [172] (Table 1).

### 3.3. Dendritic Cell-Targeting Therapies

Adoptive immunotherapy and DC-based vaccines have been created in order to reestablish an effective antitumor response by enhancing the DC/CD8+ T-cell interaction. A meta-analysis of clinical studies found that DC-based immunotherapies indicated improved outcomes, increased the CD4+ T/CD8+ T ratio, and were safe for participants [173]. Activation of DC by OK432, a streptococcus-derived anticancer immune-therapeutic drug, through CD40/CD40L ligands induced secretion of substantial quantities of Th1 cytokines (IL-12 and IFN-γ) and improved cytotoxic T-cell activity [174]. These DCs-OK432 can be generated from collected patient monocytes that have been activated with IL-4 and granulocyte-macrophage colony-stimulating factors (GM-CSF) before being treated with OK432. In a small study, HCC patients who received DCs-OK432 in parallel with embolization had longer recurrence-free survival [175].

The combined use of DC vaccination and immune-checkpoint inhibitors (ICIs) has recently demonstrated encouraging outcomes [176]. DCs expressing alpha-fetoprotein(aFP) induced a substantial decrease in proliferation and a minor delay in tumor development in pre-established in vivo subcutaneous and orthotopic HCCs, whereas a combination of aFP- and CD40L-expressing DCs had a synergistic impact related to an increased Th1-cytokine level, tumor invasion by cytotoxic T lymphocytes, and tumor apoptosis [177]. In addition, in an orthotopic HCC mouse model, the combined vaccination with DCs and treatment with a PD-1 inhibitor had a longer overall survival and more effective reduction in tumor volume compared to monotherapy with the PD-1 inhibitor [178] (Table 1).

### 3.4. T-Cell-Targeting Therapies

CD8+ T-cell responses to particular TAAs are thought to represent potential immunological antitumor factors; however, they are not highly enhanced in HCC, implying poor induction and limited antigen recognition [179]. Exhausted TAA-specific CD8+ T-cells have been observed among patients with HCC, along with an increase in activation markers in spite of poor amounts of granzyme B and effector cytokines [180]. TAA-specific T-cell responses are related to a decline in MDSCs and a reduction in HCC recurrence among individuals receiving radiofrequency ablation [181]. aFP is a widely investigated TAA in HCCs since CTL epitopes for aFP were discovered early in carcinogenesis. aFP is transcriptionally reactivated and strongly expressed in 75% of HCC patients, and high blood levels are related to poor outcomes [182] (Table 1).

The use of microwave ablation in HCC patients recently demonstrated de novo or increased tumor-specific immune responses in 30% of patients by increasing TAA [183]. Furthermore, HCC-TAAs, such as GPC3 and AFP, are being investigated as CAR-T targets. In a patient with end-stage HCC, autologous HBV-specific CAR-T-cells were able to target HBsAg-expressing HCC cells without exacerbating hepatic inflammation or damage [184]. GPC3-CAR-T cells have been shown to be capable of eliminating GPC3+ HCC cells and tumors in a patient-derived xenograft [185] and in a phase I clinical study demonstrated safe and potential effectiveness in patients with advanced GPC3+ HCC [186]. Furthermore, second and third generations of these CAR-T cells were created by disrupting PD-1 using CRISPR/Cas9 or by co-expressing the co-stimulatory molecule ICOSL-41BB [187,188].

## 4. Conclusions

HCC tumors are characterized by a complex environment characterized by interactions between tumor cells and other cell types. The TME is made up of a variety of immune cells, CAFs, and endothelial cells that provide growth factors to cancer cells while also enabling proliferation, immune evasion, and angiogenesis. Recent studies have revealed that among HCC patients, the liver TME displays a more consistent pattern than the HCC tumor cells, implying that targeting the TME rather than the tumor cells alone may be a more effective approach, bypassing tumor heterogeneity and variety. This idea is strengthened further by the fact that the observed noticeable mutations and proliferation pathways in HCC are very diverse. The recent immunotherapy treatments for HCC have shown better efficacy and less side effects compared to TKIs, but they have some restrictions and they are not effective in a substantial number of patients. Understanding the association between oncogenic pathways and immune responses is crucial in this setting for enhancing the efficacy of present and future therapies. Drug resistance appears to be primarily assigned to CSCs, which are responsible for much of the intratumor variability within every tumor population. Integrating CSC assets into our knowledge of drug resistance is critical because it may not only enable an improved comprehension of the mechanisms of chemoresistance but also promote the detection of possible biomarkers to predict the outcome of therapy along with druggable targets in order to create novel pharmacological strategies for enhancing HCC sensitivity to anticancer drugs. Furthermore, knowing the interactions between various immune cells with themselves and stromal cells, such as HSCs or CAFs, will be crucial for therapeutically exploiting the TME. The tumor microenvironment has a controversial role in the progression of hepatocellular carcinoma with various and intricate features, and therefore further in-depth studies are mandatory.

## Figures and Tables

**Figure 1 ijms-24-11471-f001:**
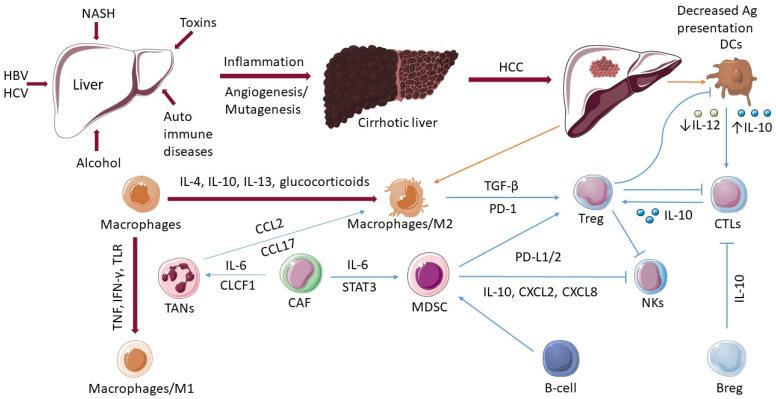
The role of immune cells in hepatocellular cancer. In the development of HCC, various factors play a decisive role such as hepatitis B virus (HBV), hepatitis C virus (HCV), non-alcoholic fatty liver disease (NAFLD), non-alcoholic steatohepatitis (NASH), toxins, and autoimmune diseases. The actions of numerous immune cells become dysregulated as the disease develops from cirrhosis of the liver to hepatocellular carcinoma (HCC). The ability of dendritic cells (DCs) to present antigens is decreased. The activation of CD8+ T-cells (CTLs) is dependent on the activation of a DCs. Macrophages differentiate into an “alternatively-activated phenotype”, M2, which promotes the recruitment and growth of regulatory T-cells (Tregs). Tregs have a negative impact on CTLs, DCs, and natural killer (NK) cells. M1 macrophages are activated by microbial components or proinflammatory cytokines (TNF, IFN-γ, TLR). Tumor-associated neutrophils (TANs) produce the chemokines, and CCL2 and CCL17 were observed to recruit tumor-associated macrophages (TAMs). Cancer-associated fibroblasts (CAFs) induce activation of TANs via IL-6 and CLCF1 and promote the differentiation of monocytes into MDSCs via IL-6-mediated STAT3 activation. B cells can be associated with pro-tumorigenic processes by activating MDSCs. Regulatory B-cells (Bregs) promote T-cell dysregulation through an IL10-dependent mechanism.

**Table 1 ijms-24-11471-t001:** Clinical trial drugs targeting cells in hepatocellular carcinoma.

Class of Target	Agent	Mechanism	Strategy of Clinical Study	Clinical Stage	Clinical Trial Number
γδ T-cells	zoledronic acid	Antiangiogenesis and antiproliferation (target tyrosine-kinase receptors and γδ T-cells)	Combination with sorafenib (TKI)	Phase II	NCT01259193
Anti-CSF-1R mAb	cabiralizumab	Repressing the activity of CSF1R-dependent TAMs	Combination with nivolumab (anti-PD1 mAb)	Phase II	NCT04050462
Pan-PI3K Inhibitor	SF1126	Reprogramming (disrupts two key MYC-mediating factors)	Combination with nivolumab (anti-PD-1 mAb)	Phase I	NCT03059147
CAR-M	pembrolizumab	Restoring phagocytic capacity	Single agent	Phase I	NCT04660929
FGFR	JNJ-42756493	Promotes ECM depletionPrevents CAF activation	Single agent	Phase-I and phase-II trials ongoing	NCT02421185
TGF-β	LY2157299	Prevents CAF activation and immunosuppression	Combination with sorafenib	Phase II	NCT02178358
FAK	dabrafenib mesylate	Reduces downstream signaling of integrins	Single agent	Phase II	NCT02465060
C-RAF, B-RAF	sorafenib	Involves in RAF/MEK/ERK pathwayVEGFR/PDGFR	Single agent	Phase II	NCT00044512
CCR2/5	BMS-813160	Cut CCL2-CCR2 and CCL5-CCR5 axis, reduce macrophage recruitment	Combination with neoadjuvant nivolumab	Phase II	NCT04123379
CSF1R, VEGFR2	regorafenib	Inhibition of CSF1R and VEGFR2	Combination with neoadjuvant nivolumab	Phase II	NCT04170556
CSF1R, VEGFR2	chiauranib	Inhibition of CSF1R and VEGFR2	Single agent	Phase II	NCT03245190
TLR7	RO7119929	Inducing macrophage repolarization	Combination with tocilizumab	Phase I	NCT04338685

## Data Availability

Not applicable.

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
