# Peer review of "Immune System and Hepatocellular Carcinoma (HCC): New Insights into HCC Progression"

_ijms, 2023, doi:10.3390/ijms241411471_

Round 1

Reviewer 1 Report

This is an interesting review addressing the major role of TME components in hepatocellular carcinoma and describes the immune system's participation in disease progression. 

The topic is of current prominent importance due to the potential impact on the emerging immunotherapies in HCC. I have some points to suggest to further improve the clinical significance of this review.

Discussing the critical immunosuppressive action of tumor microenvironment, the authors discuss the role of CD4+ CD25+ Foxp3 regulatory T cells (Tregs). They describe their role also in figure 1.

However, the role of Tregs in chronic liver diseases is different according to the etiology of liver diseases. In particular, while in viral (HBV, HCV) chronic liver diseases they are more active and expanded and this sustains the chronic infection, in autoimmune liver diseases (autoimmune hepatitis and primary biliary cholangitis) they are classically reduced both in number and function. This difference between viral and autoimmune liver disease might explain the lower prevalence of HCC in autoimmune liver disease compared to viral chronic liver disease, as recently well summarized in a comprehensive review addressing the role of Tregs in viral and autoimmune liver diseases (Hepatocellular carcinoma in viral and autoimmune liver diseases: Role of CD4+ CD25+ Foxp3+ regulatory T cells in the immune microenvironment. World J Gastroenterol. 2021 Jun 14;27(22):2994-3009. doi: 10.3748/wjg.v27.i22.2994).

-another important point is the immunomodulatory role of antiangiogenic agents in HCC. It has been recently reported that some tyrosine kinase inhibitors used in HCC treatments have immunomodulatory effects. In particular, as recently well-described (Experience with regorafenib in the treatment of hepatocellular carcinoma. Therap Adv Gastroenterol. 2021 May 28;14:17562848211016959. doi: 10.1177/17562848211016959) Regorafenib has shown anti immunosuppressive properties, as well as antitumor immunity promotion by modulating macrophages and increasing proliferation and activation of CD8 + T cells, and Cabozantinib has a synergistic effect with immune checkpoint inhibitors by acting on TAMs and decreasing tumor vascularity.

Therefore, new emerging pieces of evidence are supporting the rationale for combination therapy TKI plus ICI as recently reported (TKIs in combination with immunotherapy for hepatocellular carcinoma. Expert Rev Anticancer Ther. 2023 Mar;23(3):279-291. doi: 10.1080/14737140.2023.2181162. ).

minor editing

Author Response

Dear Reviewer.

Thank you very much for your time and effort reviewing our manuscript. We have tried to address your comments to our best. Please find attached our aswers.

Best regards

A. Armakolas

Reviewer 2 Report

In the manuscript” Immune system and hepatocellular carcinoma (HCC): New in 2 sights in HCC progression”, the authors summarized the HCC immune microenvironments and current therapeutic strategies.

In general, the manuscript is well-written. The language is overall clear and professional. The figures are well-made and properly labeled. 

However, summarization is oversimplified and lacks details. The authors described the involvement of different immune cells in HCC progression; however, the authors did not fully cover the key point: how HCC cells, especially cancer stem cells, escape immune surveillance. This could be more important and valuable for clinical applications.  

Here are the major points the author needs to revise:

1.  The authors should have a chapter to discuss the impact of NK cells in the HCC microenvironment.

2.  In chapter/part 2, the authors named the chapter/part “microenvironment in cancer”. The authors should have discussed the role of TAF as TAF is a major microenvironment component that interacts with both tumor cells and immune cells.

3. Chapter/part3 is oversimplified. The authors should have an extended and in-depth discussion on this chapter and have a table or chart to summarize these therapeutic strategies, their target, pre-clinical/clinical stages, and potential mechanism.

4. The major challenge in immune therapy is immune evasion and resistance, both are predominantly contributed by cancer stem cells. The authors should have a discussion regarding the mechanism of immune invasion and drug resistance in cancer stem cells. 

Minor:

1. Line 70-71, the text is in italics.

Author Response

Dear Reviewer 

Thank you very much for you time and effort revising our manuscript. We have tryied our best to answer your comments. Please find attached our answers.

Best regards

A. Armakolas
